# DecipherPref: Analyzing Influential Factors in Human Preference Judgments via GPT-4

**Yebowen Hu,**[†] **Kaiqiang Song,**[‡] **Sangwoo Cho,**[‡] **Xiaoyang Wang,**[‡] **Hassan Foroosh,**[†] **Fei Liu**[§]

[†]University of Central Florida  [‡]Tencent AI Lab, Bellevue, WA  [§]Emory University
{yebowen.hu,hassan.foroosh}@ucf.edu
{riversong,swcho,shawnxywang}@global.tencent.com  fei.liu@emory.edu

## Abstract

Human preference judgments are pivotal in guiding large language models (LLMs) to produce outputs that align with human values. Human evaluations are also used in summarization tasks to compare outputs from various systems, complementing existing automatic metrics. Despite their significance, however, there has been limited research probing these pairwise or $k$-wise comparisons. The collective impact and relative importance of factors such as output length, informativeness, fluency, and factual consistency are still not well understood. It is also unclear if there are other hidden factors influencing human judgments. In this paper, we conduct an in-depth examination of a collection of pairwise human judgments released by OpenAI. Utilizing the Bradley-Terry-Luce (BTL) model, we reveal the inherent preferences embedded in these human judgments. We find that the most favored factors vary across tasks and genres, whereas the least favored factors tend to be consistent, e.g., outputs are too brief, contain excessive off-focus content or hallucinated facts. Our findings have implications on the construction of balanced datasets in human preference evaluations, which is a crucial step in shaping the behaviors of future LLMs.

## 1 Introduction

Human judgments are widely used in summarization evaluation, complementing automatic metrics. Automatic metrics often rely on reference texts. They compare system summaries with reference texts based on word overlap, as seen in BLEU and ROUGE (Papineni et al., 2002; Lin, 2004), or contextualized embeddings, as in BERTScore, MoverScore, BLEURT, UniEval (Zhang et al., 2019; Zhao et al., 2019; Sellam et al., 2020; Zhong et al., 2022). By contrast, human evaluation usually does not require reference texts. Evaluators are asked to assess the quality of summaries along certain linguistic and content dimensions; they can also rank outputs from various systems (Bhandari et al., 2020;

Fabbri et al., 2021; Clark et al., 2021). This type of reference-free evaluation has proven beneficial, especially as systems are increasingly matching the performance of humans.

Pairwise human judgments have become extensively used in recent years for the development of large language models (Ziegler et al., 2020; Stiennon et al., 2020; Nakano et al., 2022; Ouyang et al., 2022; Menick et al., 2022; Bai et al., 2022; Ramamurthy et al., 2023). InstructGPT, for example, learns a reward model from human comparisons, then optimizes against this reward model to generate outputs that align with human preferences.[1] Human ratings are further used in the final evaluations to rank model variants based on specific prompts and their completions (Ouyang et al., 2022). Models with higher winrates are considered to have an edge. Human comparisons have been chosen for their ease and intuitiveness, and they play a critical role in shaping the behavior of LLMs.

Despite their significance, there has been limited research probing human preference judgments. Using summarization as a case study, we examine the characteristics of system outputs that may influence human judgments. We have selected summarization as our case study, because summaries need to be anchored in their original texts, which facilitates our evaluation of content accuracy. It is worth noting that our proposed framework could potentially be extended to other scenarios. Evaluators of summarization and text generation systems are typically asked to consider a range of factors: informativeness, factuality, fluency, coherence, extractiveness, non-redundancy, *etc.* (Howcroft et al., 2020; Fabbri et al., 2022; Goyal et al., 2022; Zhang et al., 2023). However, the aggregate effect and relative importance of these factors remain elusive. E.g.,

---

[1]Beyond comparing two model outputs for the same input (pairwise comparisons), evaluators use K-wise comparisons for efficiency. These K-wise comparisons are later converted into $\binom{K}{2}$ pairwise comparisons for training reward models.

factual errors can greatly undermine a system's output. An inaccurate claim made by Google's Bard about the James Webb Space Telescope has resulted in a 7% drop in stock price (Vincent, 2023). We hypothesize that there may be other, uncovered factors influencing human judgments.

In this paper, we examine a dataset of pairwise human judgments released by OpenAI (Stiennon et al., 2020). Much like a consumer weighing up multiple factors when deciding between two products, we aim to uncover the factors that evaluators consider when assessing two system outputs. We leverage the Bradley-Terry-Luce model to analyze the strengths of these factors (Bradley and Terry, 1952), which has found applications in psychology, marketing, economics, and evaluation of natural language systems (Dras, 2015; Zopf, 2018; Peyrard et al., 2017, 2021; Sedoc and Ungar, 2020). It models pairwise comparisons, such as sports matches or political elections, by assuming each factor has an inherent strength, and the likelihood of one factor winning over another is a function of the difference between their strengths. We develop a protocol to identify characteristics of system outputs, ranging from output length and content coverage, to hallucinations of various sorts and the usage of complex words. Our research sheds light on inherent preferences embedded in human judgments.

Our paper's contributions include: (a) A comprehensive analysis of a collection of human comparisons to identify key factors that may influence human judgments. In this analysis, we use summarization as a case study with comparisons provided by OpenAI. (b) Using GPT-4's advanced capabilities, we assess system outputs both qualitatively and quantitatively. We examine their fluency, clarity, coverage, alignment with the original text's intent and style, and detect hallucinations based on atomic facts (Liu et al., 2023c). Our study of influential factors holds promise for enhancing the reliability of human evaluations.[2]

## 2 Problem Formulation

Our dataset consists of a collection of $N$ summary pairs, denoted by $\mathcal{D} = \{(S_1^{(n)}, S_2^{(n)})\}_{n=1}^N$. The two summaries, generated for the same input, are accompanied by the original text and a human judgment of their overall quality, with $\hat{S}^{(n)}$ denoting the favored summary of the $n$-th pair. We develop

---

[2]Our data and analyses are publicly available for the research community: https://decipherpref.github.io/

Figure 1: TOP: A prompt for GPT-4 to extract atomic content units from a summary. BOTTOM: A prompt to determine whether an atomic content unit contains any location, temporal, possessive expressions, or quantities.

a set of $M$ factors, represented as $\mathcal{A} = \{a_i\}_{i=1}^M$, to examine each summary from multiple perspectives. We consider their style, intent, richness of content, connection to the original text, *etc*. Some of these factors, such as style, use of complex words, have not been explicitly provided to the evaluators in the instructions (see §3 for further details). We use $\mathbf{a}_1^{(n)}$ and $\mathbf{a}_2^{(n)}$ to represent the unique factors identified for the two summaries, where $P_n$ and $Q_n$ are the number of factors (Eq. (1-2)). The goal of this study is to identify dominating factors that influence human judgments.

$$\mathbf{a}_1^{(n)} = \{a_{1,p}^{(n)}\}_{p=1}^{P_n}, \quad a_{1,p}^{(n)} \in \mathcal{A} \qquad (1)$$

$$\mathbf{a}_2^{(n)} = \{a_{2,q}^{(n)}\}_{q=1}^{Q_n}, \quad a_{2,q}^{(n)} \in \mathcal{A} \qquad (2)$$

We use the Bradley-Terry-Luce model ([Bradley and Terry, 1952](#)) to rank factors that characterize system outputs. The BTL model is frequently used in sports to rank players within a league. Suppose we have $M$ players who have participated in a number of games against each other. We can represent the outcomes of these games using a matrix $W$; $w_{i,j}$ $(i, j \in M)$ denotes the number of times Player $i$ has won over $j$. If $i$ has never competed against $j$, we assign $w_{i,j}$ to 0. Further, a player cannot compete against themselves, so $w_{i,i}$ $(i \in M)$ is also set to 0. $W_i$ is the total wins that Player $i$ has had against all other players. Given this matrix $W$, we can calculate the relative strengths of all players using the Bradley-Terry-Luce model.

We treat each unique factor as a player and aim to understand the strengths of these factors based on pairwise comparisons. When one summary is favored over another, i.e., $S_1^{(n)} \succ S_2^{(n)}$, we assume all of its factors win over those of the other summary: $a_{1,p}^{(n)} \succ a_{2,q}^{(n)}, \forall p, q$, and vice versa.[3] If the same factor appears in both summaries, we assume they cancel each other out and exclude them from the list of factors. This results in $\{a_{1,p}^{(n)}\}_{p=1}^{P_n}$ and $\{a_{2,q}^{(n)}\}_{q=1}^{Q_n}$ being the symmetric difference of the two summaries. Thus, we have $P_n \times Q_n$ factor comparisons derived from each pair of summaries. The BTL model allows us to estimate the relative importance of these factors, represented as $\{p_i\}_{i=1}^M$. This is accomplished using an EM-like algorithm, where $p_i$ is iteratively updated (Eq. (3)) to maximize the data likelihood and then renormalized (Eq. (4)). The BTL model is primary used for parameter estimation in pairwise comparisons ([Zhu et al., 2023](#)), making it an ideal fit for our dataset.

$$p_i' = W_i \left( \sum_{j \neq i} \frac{w_{ij} + w_{ji}}{p_i + p_j} \right)^{-1} \qquad (3)$$

$$p_i = \frac{p_i'}{\sum_{j=1}^M p_j'} \qquad (4)$$

## 3 Determinants of Human Preferences

We explore a variety of factors that may influence human preferences. These factors are high-level, interpretable descriptors of the summaries and their original texts. Our work distinguishes from previous studies using LLMs to evaluate the quality of

---

[3]This assumption, like the conditional independence in Naive Bayes, is quite strong. However, the BTL model generally performs well and shows relative resilience to its violation.

---

Figure 2: TOP: A prompt for GPT-4 to verify whether each atomic content unit accurately matches the original information, helping us detect hallucinations. BOTTOM: A prompt to check if the atomic content unit is related to the main focus of the original text, which helped us detect off-focus content.

summaries, which models summaries and original texts based on low-level contextualized representations ([Luo et al., 2023](#); [Liu et al., 2023a](#); [Gao et al., 2023](#)). Our objective in this paper is to uncover the inherent preferences in pairwise human judgments. By doing so, we aim to establish a robust sample collection practice for reward models.

***Length.*** Previous research suggests that human evaluators might show a bias towards lengthier summaries ([Liu et al., 2023c](#)). Indeed, a longer summary can seem more credible due to its comprehensiveness. Conversely, a short summary may miss some critical content due to its brevity. Evaluators may naturally choose the lengthier output

Figure 3: Two prompts for GPT-4 to assess whether a given summary is fluent (TOP) or clear (BOTTOM).

Figure 4: Two prompts for GPT-4 to check if the given summary aligns with the original text in terms of style (TOP) and intent (BOTTOM).

without verifying its content. Despite this, there has been little effort to quantify the influence of summary length. In this study, we measure the length of summaries by counting the number of tokens or characters, then categorize all summaries into quartiles based on their length. E.g., the factor `len-tk-medium` indicates the summary's length falls into the second quartile when measured by tokens. We have the following length factors:

▶ `len-{tk|ch}-{short|medium|long|xlong}`

***Linguistic Quality.*** Assessing linguistic quality with a 5 or 7-point Likert scale can be ineffective due to subjective interpretation (Howcroft et al., 2020). This issue is often exacerbated by the lack of clear guidelines for evaluators. Our approach models each linguistic quality as a binary decision, thus enabling a clear judgement on each summary. We measure (a) `fluency`: whether the summary is easy to understand and free of grammatical errors; (b) `clarity`: if the summary expresses ideas clearly and unambiguously.[4]

A summary's style and intent, as it relates to the original text, can also impact human perception. We evaluate (c) `style-alignment`: if the summary is written in the same style as the original text, e.g., formal, casual, humorous, sarcastic, etc., and (d) `intent-alignment`: if the summary serves the same purpose as the original text, e.g., soliciting advice, sharing information, etc. We derive four factors leveraging the exceptional capabilities of

GPT-4. Our instructions are adapted from Stiennon et al., (2020) and illustrated in Figures 3 and 4.

▶ `{style|intent}-aligned`
▶ `{fluent|unambiguous}`

***Content Accuracy.*** Evaluators should disapprove summaries that contain hallucinated content. We measure this aspect by counting the number of hallucinated Atomic Content Units (ACUs). An ACU is a self-contained information unit that does not require further breakdown, such as '*Raheem Sterling has not signed a new Liverpool contract*.' Using expert-annotated ACUs provided by Liu et al.(2022) as in-context examples, we employ GPT-4 to extract atomic facts from a given summary, as illustrated in Figure 1.

We determine if the factual information within each ACU aligns accurately with the original text using GPT-4; our prompt is shown in Figure 2. An ACU is deemed accurate if it neither introduces unmentioned details nor contradicts the original text. We sort all summaries into four categories based on the number of fabricated ACUs they contain: none, one, two, or more. The underlying assumption is that a summary should include minimal, in any at all, hallucinations. In addition, fabricated elements, such as location, time and possessive expressions, and quantities can lead to major misinterpretations. Thus, we define a 'hallucination-mixed-type' factor to flag any summary that includes at least one

---

[4]Given that text coherence focuses on the fluidity of content at the level of paragraphs or larger sections, we decided to exclude it from our criteria. This decision helps mitigate overlap with fluency and accounts for the brevity of summaries.

hallucinated ACU that contains these expressions.

▶ `hallucination-fact-{none|one|two|many}`
▶ `hallucination-mixed-types`

***Connection to the Original Text.*** Does the source content coverage impact human judgments? To answer this, we calculate a 'coverage score', which represents the ratio of reused content to the total content in the source text. We compute reused content using Grusky et al.'s (2018) greedy algorithm that aligns text fragments of a summary with the original text. We also consider whether the summary mostly reuse individual words or consecutive fragments from the source text. This is quantified using their 'density score', which favors consecutive reused snippets by squaring their lengths.

It's important to note that these methods assess the extent of coverage, not necessarily the importance of the covered content. We quantify the total number of 'off-focus' facts in the summary, computed as the number of ACUs that originate from the source document, but may not represent the main content. We leverage GPT-4 to evaluate if any atomic fact is related to the main focus of the original text, where the main focus is the core subject around which all the content revolves (Table 2).

▶ `off-focus-{none|one|two|many}`
▶ `src-cov-{minimal|low|medium|high}`
▶ `consec-cov-{minimal|low|medium|high}`

***Word Choice.*** We use the algorithm introduced by Grusky et al. (2018) to match text fragments in a summary with the original text. We term 'novel words' as the proportion of the summary that is not aligned with the original text. Further, we identify 'complex words' as those that need to be broken down into several tokens by a tokenizer. We then compute the percentage of summary tokens that are part of a complex word. Based on these measures, we categorize all summaries into four quartiles, prefixed with 'novel-words-' and 'complex-words-', respectively.

▶ `novel-words-{few|some|many|most}`
▶ `complex-words-{few|some|many|most}`

## 4  Data

In this study, we examine a large dataset released by OpenAI (Stiennon et al., 2020), which includes labeled comparisons between pairs of summaries (referred to as "`comparisons`") for the Reddit TL;DR dataset and human ratings of summaries along multiple axes ("`axis evals`") for both Reddit TL;DR

Figure 5: A prompt for GPT-4 to choose the better summary from a given pair. We noticed that the order of summary presentation and positioning of the original text affect the model's performance. For example, scenarios a) and b) do not consistently yield the same prediction, and scenarios b) and c) do not consistently produce opposite predictions. Based on a pilot study with six different combinations, we decided to use a) for our final experiments for its stability and performance.

and CNN/DM datasets. The summaries used for evaluation are generated using their human feedback models and several baselines. Our data splits are constructed in the following manner:

- **`comparisons-reddit`**. Human evaluators indicate both the better summary and their confidence in that selection using a 9-point slider. A score of 1 or 9 means that one summary is 'definitely' preferred over the other. The validation split of this dataset has around 83.8k annotated pairwise comparisons, from which we randomly selected 5k pairs for our experiments. Our initial analysis shows this distribution of pairs across confidence levels: 26% are 'possibly' better, 25% are 'likely' better, 19% are 'very likely' better, and 30% are 'definitely' better.

- **`axis-evals-{reddit|cnndm}`**: Each summary is individually assessed by an evaluator using a 7-point Likert scale. We specifically consider the overall quality of the summary (*how good is the summary overall at representing the post?*). A score of 1 means the summary is terrible while a score of 7 suggests that it is excellent. There are around 1k Reddit posts and 0.6k CNN/DM articles, each having multiple annotated summaries.

| Rank | Most Favored Factor | Estimate | | Rank | Least Favored Factor | Estimate |
|------|---------------------|----------|---|------|----------------------|----------|
| 1 | intent-aligned | .0742 | | 1 | hallucination-fact-many | .0067 |
| 2 | unambiguous | .0591 | | 2 | off-focus-many | .0100 |
| 3 | style-aligned | .0499 | | 3 | off-focus-two | .0126 |
| 4 | fluent | .0428 | | 4 | src-cov-minimal | .0130 |
| 5 | src-cov-high | .0395 | | 5 | len-ch-short | .0139 |
| 6 | off-focus-none | .0378 | | 6 | hallucination-fact-two | .0142 |
| 7 | hallucination-fact-none | .0377 | | 7 | hallucination-mixed-types | .0160 |
| 8 | len-ch-xlong | .0335 | | 8 | consec-cov-minimal | .0161 |
| 9 | len-ch-long | .0320 | | 9 | len-tk-short | .0164 |
| 10 | consec-cov-high | .0316 | | 10 | off-focus-one | .0171 |

Table 1: Most and least favored factors in our `comparisons-reddit` dataset identified by the BTL model.

We pair these summaries and randomly pick two pairs per document, ensuring that they have different ratings for overall quality. This process gives us a total of 2,058 Reddit TL;DR summary pairs and 1,254 CNN/DM pairs.[5]

## 5 Experiments

In this section, we discuss various factors that affect human preferences. We quantitatively measure their degree of influence, how accurately these factors can be extracted from pairs of summaries, how factors correlate with one another, and their varying effects across summarization tasks and source domains. Further, we evaluate how top-performing GPT models fare on this task. To ensure the replicability of our results, we set the temperature to 0 when using all GPT models.

### 5.1 Factors That Influence Human Preference

Utilizing the BTL model (§2), we identify the most and least preferred factors on our 'comparisons-reddit' dataset; results are shown in Table 1. Additional results for 'axes-evals-{reddit|cnndm}' can be found in Tables 5 and 6 in the supplementary materials. We make the following observations:

**Hallucination & Focus**. Not having any hallucinations ('hallucination-fact-none') is not the most influential factor on human preferences. However, too much hallucination certainly harms system outputs. This is seen with 'hallucination-fact-many',

which is rated as the worst among all evaluated. The 'hallucination-fact-two' and 'hallucination-mixed-types' factors also rank among the least favored. Likewise, we observe that the inclusion of irrelevant content in the summary is not preferred. In particular, including two or more off-focus atomic facts has a negative impact on the system outputs.

**Linguistic Quality**. Interestingly, our research indicates that human evaluators show a strong preference for certain linguistic aspects of summaries. Particularly, 'intent-aligned' consistently ranks the highest in both datasets: `comparisons-reddit` and `axes-evals-reddit`. This suggests that maintaining the original intent of the post, such as seeking advice, sharing information, asking questions, or providing support, is of importance to a Reddit summary. It is also crucial that the summary is fluent, maintains the original post's style, and expresses ideas clearly and without ambiguity.

**Length**. Our findings show that human evaluators tend to favor longer news summaries while showing a dislike for shorter ones, as seen in Table 6. This is because longer summaries can provide more details about the news stories. However, this preference for length is not as obvious with Reddit summaries. Particularly, it is not a case of 'the longer, the better'. Summaries that are excessively long, denoted as 'len-ch-xlong,' only ranks 8th in terms of preferred factors. On the flip side, summaries that are overly short, denoted as 'len-ch-short,' can negatively affect human preferences. As a result, our recommendation is to tailor output length to the specific task, with a greater emphasis on length for news summarization.

**Miscellaneous**. We observe that system summaries tend to suffer when there is minimal coverage of the source content, judged by either individual words (src-cov-minimal) or consecutive chunks

---

[5]An alternate approach to gathering pairwise data is pairing system outputs with reference texts. Summaries written by editors have traditionally served as reference texts in news summarization. However, recent research indicates that LLMs have achieved human-level performance on news datasets such as XSum and CNN/DM (Zhang et al., 2023). As a result, assuming system outputs are inferior to reference texts without further human evaluation is no longer valid. We leave automatic collection of pairwise data for future work and focus on human judgments provided by OpenAI in this study.

| Confidence Level | Pairs | GPT Models | | |
|---|---|---|---|---|
| | | davinci-003 | 3.5-turbo | gpt-4 |
| Summary 1 is *possibly* better than Summary 2 | 1,273 | 53.54 | 56.71 | 61.01 |
| Summary 1 is *likely* better than Summary 2 | 1,227 | 60.81 | 65.15 | 72.63 |
| Summary 1 is *very likely* better than Summary 2 | 964 | 59.25 | 66.84 | 74.74 |
| Summary 1 is *definitely* better than Summary 2 | 1,536 | 67.01 | 74.26 | 81.46 |
| All Pairs Combined | 5,000 | 60.56 | 66.15 | **72.79** |

Table 2: Accuracy of various GPT models when selecting the better summary from a given pair, as evaluated on the `comparisons-reddit` dataset.

| Gap | Pairs | GPT Models | | |
|---|---|---|---|---|
| | | davinci-003 | 3.5-turbo | gpt-4 |
| 1 | 802 | 56.25 | 65.98 | 68.82 |
| 2 | 576 | 66.26 | 72.61 | 80.87 |
| 3 | 358 | 71.42 | 82.67 | 87.71 |
| 4 | 179 | 69.83 | 87.50 | 92.13 |
| 5 | 101 | 85.00 | 94.95 | 98.02 |
| 6 | 42 | 82.93 | 97.61 | 100.00 |
| All | 2,058 | 64.81 | 74.70 | **79.57** |

Table 3: Accuracy of various GPT models when selecting the better summary from a given pair, as evaluated on the `axis-eval-reddit` dataset. **Gap** refers to the difference in overall scores between two summaries.

of text (consec-cov-minimal). This pattern holds true across all datasets. Factors that do not fall into either the most favored or least favored categories include: medium length, a single hallucinated fact or off-focus content unit, moderate source content coverage, and nearly all factors tied to the use of novel or complex words. Generally, these factors do not significantly enhance or harm the quality of the summaries.[6]

***Understanding Factor Correlations.*** The interplay among factors can be complex, partly due to the complicated nature of human decision-making. We use Kendall's Tau to measure the relationships between these factors. Results are shown in Figure 7 in the supplementary material. We observe that, factors within each category, such as hallucination-fact-{none, one, two, many}, show negative correlations. This is because for each system output,

only one of these factors will be activated. Moreover, there is notable correlation between 'short output length' and 'minimal source content coverage,' as short summaries often leave out salient source content due to brevity. Preferred linguistic traits, i.e., 'fluent', 'unambiguous', 'style-aligned' and 'intent-aligned', correlate positively with high content quality ('off-focus-none', 'hallucination-none'). They are negatively correlated with the use of new and complex words, multiple hallucinated facts or off-focus content. This suggests that better language and content qualities are likely attained together when LLMs' abilities are improved. We believe there might be complex three-way interactions among these factors, further complicating the preference landscape.

***Accuracy of Factor Extraction.*** We use GPT-4 to extract atomic content units (ACUs) from summaries for our analysis. To understand the efficacy of GPT-4, we randomly select 50 summaries for manual validation. Our findings indicate that GPT-4 accurately processed 43 out of 50 summaries, yielding an accuracy of 86%. The majority of error instances are due to the informal nature of Reddit posts. For example, "*What do?*" is a casual expression for "*What should I do?*", seeking advice on a specific situation. GPT-4 has failed to extract an ACU from it. We believe that parameter-efficient fine-tuning might enhance LLMs' ability to process such informal text (Li et al., 2022, 2023b). GPT-4 achieves an accuracy of 89% on detecting hallucination instances and 83% in checking if an ACU was relevant to the main focus of the original text.

***Cross-Source Analysis.*** We observe that factors influencing summarization tasks can differ across varying source domains. Figure 6 shows a comparison on the Reddit and CNN/DM. For news summarization, summaries that are longer and more

---

[6]We test the BTL model's robustness using nonsensical factors. Our findings indicate that when random factors are introduced, the BTL model does not rank them among the most or least favored, suggesting the model is robust.

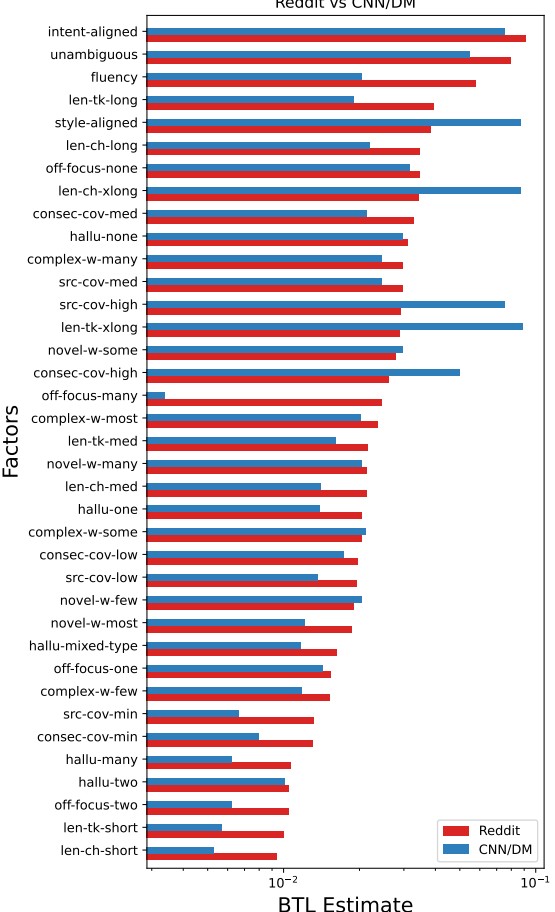

Figure 6: Factors impacting summarization tasks can vary across different source domains. For news summarization, which is shown in blue, lengthier summaries are often favored (`len-{ch|tk}-xlong`), as is comprehensive coverage of the source content (`{src|consec}-cov-high`). The style of the summary reflects that of the original article (`style-aligned`), and off-focus content tends to have minimal impact (`off-focus-many`).

comprehensive tend to be preferred, since they provide a more detailed coverage of the source content. Off-focus content appears to have a minimal impact, as ACUs extracted from summaries usually maintain relevance to the focus of the news article.

### 5.2 Results of Pairwise Judgments

The accuracy of various GPT models on pairwise judgments is shown in Tables 2 and 3. We use `text-davinci-003`, `gpt-3.5-turbo-0301`, and `gpt-4-0314` for this study. In particular, `text-davinci-003` allows instruction-following. `gpt-3.5-turbo` is optimized for chat and performs well in text completion tasks. `gpt-4` is the latest model made available through OpenAI's API.

We divided our data into multiple splits based on the confidence level evaluators have assigned. In

the `comparisons-reddit` dataset, there are four splits labeled as Summary 1 being {definitely, very likely, likely, possibly} better than Summary 2. Similarly, the `axis-evals-reddit` dataset has six splits based on the gap of overall scores. A Gap of 1 represents minimal difference in overall quality, while a Gap of 6 indicates the maximum difference. In all cases, GPT-4 consistently outperformed other models regardless of the dataset. However, when two summaries had similar quality levels, identifying the better one was quite challenging. In this category, the best-performing GPT-4 model achieved accuracies of 61.01% and 68.82% for the two datasets, respectively.[7]

## 6 Related Work

***Evaluation of LLMs.*** Human evaluation of LLMs and other NLP systems has become more important than ever, as these systems are prone to inherent biases and may generate hallucinated facts (Maynez et al., 2020; Lebanoff et al., 2020a,b; Liang et al., 2022; Cao et al., 2022; Laban et al., 2023). Pairwise preference judgments are often selected for their simplicity and intuitiveness. They require less cognitive effort than rating individual model outputs on a Likert scale (Dras, 2015; Perez-Ortiz and Mantiuk, 2017). In recent years, GPT-4 has been employed as a surrogate for human evaluators to conduct these preference judgments (Chiang et al., 2023; Liu et al., 2023a,b). The goal of our study is to identify the key factors derived from pairwise comparisons to enhance the transparency of human preference judgments.

***Human-AI Alignment.*** LLMs are often adapted to encourage preferred model behaviors and discourage undesired ones based on a learned reward function, a process often known as alignment (Köpf et al., 2023; Wolf et al., 2023). Previous research has explored various reward functions to guide language generation. These range from maximizing ROUGE scores to using question answering-based rewards (Pasunuru and Bansal, 2018; Peyrard and Gurevych, 2018; Arumae and Liu, 2019; Laban et al., 2020; Yadav et al., 2021; Yu et al., 2022). Additionally, human feedback for LLMs can manifest in various forms, including natural language-based and more fine-grained feedback (Scheurer et al., 2022; Wu et al., 2023; Lee et al., 2023). Our

---

[7]We anticipate that enabling GPT-4 to reason over salient factors could potentially enhance its ability to judge the quality of system outputs. However, this is beyond the current study.

research into the influential factors that affect human preferences can provide valuable insights for reward factorization. Such factor analysis may also help researchers mitigate potential biases in the alignment between humans and AI.

***Preference Modeling.*** Preference modeling allows us to incorporate arbitrary prior knowledge about users into the learning process in a declarative way (Lu and Roth, 2012). It is critical for the output from LLMs to respond to a variety of user attributes, ranging from interaction history to the situation of use. In the past, researchers have investigated user modeling for various tasks, including headline generation, dialog response generation and recipe creation (Majumder et al., 2019; Flek, 2020; Wu et al., 2021; Dudy et al., 2021; Cai et al., 2023). In this paper, our focus has been on exploring human preference modeling for LLM development.

Human preferences can be influenced by various factors. In this study, preference judgments were provided by OpenAI. The demographics of evaluators could influence these judgments. OpenAI indicates that their evaluators, recruited through Upwork or Scale AI, are quite young, with 75% being under 35 years old (Ouyang et al., 2022). The gender distribution is fairly balanced, and most evaluators come from the U.S. or Southeast Asia. The researchers who provide instructions and interact with evaluators can also impact study outcomes. We believe that taking human factors into account when modeling preferences is important for future research in this domain (Li et al., 2023a).

## 7  Conclusion

We aim to uncover the factors that influence human preferences. Our research indicates that key factors are tied to users' information needs. Users expect LLMs to understand their information-seeking intents, produce precise responses without irrelevant content or hallucinations, and generate moderate-length outputs of high linguistic quality. Future research may also consider developing a balanced set of examples for human preference labeling to guide model behaviors more efficiently.

## Limitations

While the Bradley-Terry-Luce (BTL) model employed in our study is robust and widely applicable, it may not fully capture the complexities of human preference judgments. Our study did not account for the impact of cultural and demographic factors on preference judgments. We also omit considerations related to toxicity, which refers to rude, disrespectful, sexual, or violent content. These types of content are extremely rare in our dataset but could still significantly impact human preferences. Our research primarily focuses on the linguistic and content-related factors, such as output length, source content coverage, intent, fluency, and factual consistency. While these factors are identified as significant, there may be other unexplored factors that we did not consider in this study. In addition to these predefined factors, it would be interesting to automatically infer other influential factors, which we leave for future work. Finally, our findings are based on specific tasks and source domains, and caution should be taken when generalizing these results to other scenarios.

## Ethics Statement

The human preference data used in this study were provided by OpenAI. The data do not include user demographics or personally identifiable information and were released solely for research purposes. Our study aims to advance the understanding of human preferences in order to guide the development of LLMs and does not make use of technology related to user demographics. We acknowledge that our understanding of the factors influencing preferences is continually evolving as user behaviors may change over time. We emphasize the importance of constructing balanced datasets for learning reward functions. If certain aspects are disproportionately represented, the reward function may not learn effectively, potentially leading to inherent biases. We are committed to the responsible development of LLM systems, ensuring that they respect human values and that they are designed to mitigate potential biases.

## Acknowledgements

We would like to thank the reviewers for their insightful feedback, which greatly enhanced our paper. This research has been partially supported by National Science Foundation grant IIS-2303678.

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

**Summary A**

Graduated Dec. 2010 (Computer Science), can't find a tech job in MS, New Orleans, or San Francisco. Looking for tips/places to apply.

```
len-tk-medium
len-ch-medium
hallucination-fact-two
src-cov-medium
consec-cov-low
novel-words-most
complex-words-many
```

**Summary B**

I'm a recent college graduate with a degree in computer science. I have no real work experience and am desperate for a job. I'm worried that my lack of experience is hurting me.

```
len-tk-long
len-ch-xlong
hallucination-fact-one
hallucination-mixed-type
src-cov-high
consec-cov-medium
novel-words-many
complex-words-some
```

Table 4: Example summaries and their related factors. If a factor is in both summaries, we consider it neutral and remove it from the list.

# A  Example Factors

We provide an example of a pair of summaries and their related factors in Table 4.

# B  Factors That Influence Human Preference

We showcase the influential factors identified via the BTL model for the 'axes-evals-reddit' and 'axes-evals-cnndm' datasets in Tables 5 and 6 respectively. In addition, we use Kendall's Tau to evaluate the correlations among these factors. Results are shown in Figure 7.

| Rank | Most Favored Factor | Estimate |
|------|---------------------|----------|
| 1 | intent-aligned | .0915 |
| 2 | unambiguous | .0798 |
| 3 | fluent | .0577 |
| 4 | len-tk-long | .0395 |
| 5 | style-aligned | .0385 |
| 6 | len-ch-long | .0347 |
| 7 | off-focus-none | .0346 |
| 8 | len-ch-xlong | .0343 |
| 9 | consec-cov-medium | .0329 |
| 10 | hallucination-fact-none | .0310 |

| Rank | Least Favored Factor | Estimate |
|------|----------------------|----------|
| 1 | len-ch-short | .0094 |
| 2 | len-tk-short | .0101 |
| 3 | off-focus-two | .0105 |
| 4 | hallucination-fact-two | .0105 |
| 5 | hallucination-fact-many | .0107 |
| 6 | consec-cov-minimal | .0130 |
| 7 | src-cov-minimal | .0132 |
| 8 | complex-words-few | .0153 |
| 9 | off-focus-one | .0153 |
| 10 | hallucination-mixed-types | .0163 |

Table 5: Most and least favored factors in our `axis-evals-reddit` dataset identified by the BTL model.

| Rank | Most Favored Factor | Estimate |
|------|---------------------|----------|
| 1 | len-tk-xlong | .0891 |
| 2 | len-ch-xlong | .0875 |
| 3 | style-aligned | .0873 |
| 4 | src-cov-high | .0750 |
| 5 | intent-aligned | .0750 |
| 6 | unambiguous | .0549 |
| 7 | consec-cov-high | .0499 |
| 8 | off-focus-none | .0317 |
| 9 | novel-words-some | .0296 |
| 10 | hallucination-fact-none | .0296 |

| Rank | Least Favored Factor | Estimate |
|------|----------------------|----------|
| 1 | off-focus-many | .0034 |
| 2 | len-ch-short | .0053 |
| 3 | len-tk-short | .0057 |
| 4 | off-focus-two | .0062 |
| 5 | hallucination-fact-many | .0063 |
| 6 | src-cov-minimal | .0067 |
| 7 | consec-cov-minimal | .0079 |
| 8 | hallucination-fact-two | .0101 |
| 9 | hallucination-mixed-types | .0117 |
| 10 | complex-words-few | .0118 |

Table 6: Most and least favored factors in our `axis-evals-cnndm` dataset identified by the BTL model. Our findings show that human evaluators tend to favor longer news summaries while showing a dislike for shorter ones. This is because longer summaries can provide more details about the news stories. However, this preference for length is not as obvious with Reddit summaries. It is not a case of 'the longer, the better'. As a result, our recommendation is to tailor output length to the specific task, with a greater emphasis on length for news summarization.

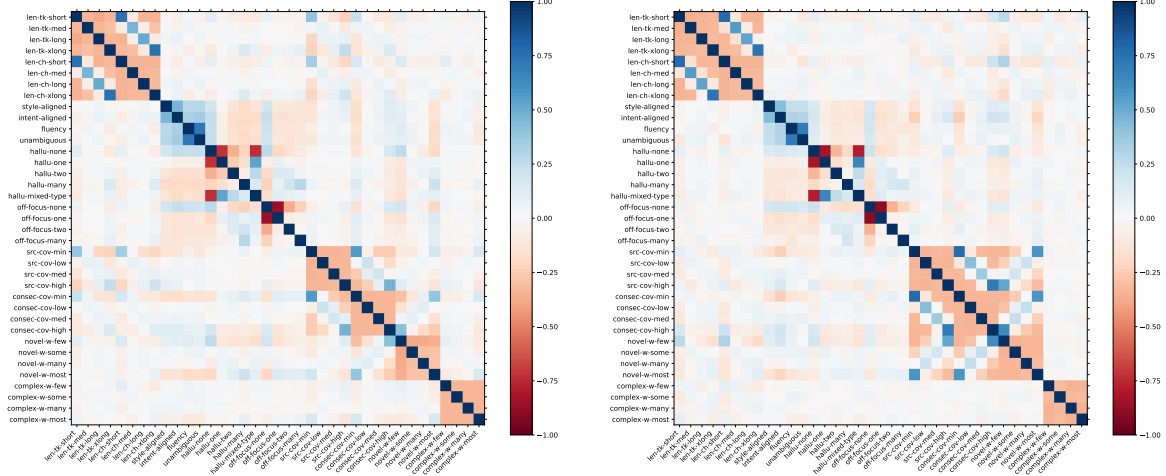

Figure 7: Factor correlations measured by Kendall's Tau. LEFT: Results obtained for `comparisons-reddit`. RIGHT: Results obtained for `axes-evals-reddit`. We observe that factors within each category, such as hallucination-fact-{none, one, two, many}, show negative correlations. This is because for each system output, only one of these factors will be activated. Moreover, there is notable correlation between 'short output length' and 'minimal source content coverage,' as short summaries often leave out salient source content due to brevity. Preferred linguistic traits, i.e., 'fluent', 'unambiguous', 'style-aligned' and 'intent-aligned', correlate positively with high content quality ('off-focus-none', 'hallucination-none'). They are negatively correlated with the use of new and complex words, multiple hallucinated facts or off-focus content. This suggests that better language and content qualities are likely attained together when LLMs' abilities are improved. We believe there might be complex three-way interactions among these factors, further complicating the preference landscape.