# OpenReview forum: "DecipherPref: Analyzing Influential Factors in Human Preference Judgments via GPT-4"
_EMNLP/2023/Conference — EMNLP 2023 Main_

### Official Review · Reviewer_ZwsG · 2023-07-21

**Soundness:** 4

**Excitement:**

4: Strong: This paper deepens the understanding of some phenomenon or lowers the barriers to an existing research direction.

**Paper Topic And Main Contributions:**

Human preference judgments act as training data for reward models, and as such are key ingredients for training more aligned language models with RLHF. The authors analyse which concrete factors (e.g. length and factuality) influence one-dimensional human preference judgments over pairs of summaries. For this purpose, the authors use a Bradley-Terry-Luce model. For Reddit and News summaries from a dataset collected by OpenAI, they find that the most favoured factors vary by task and genre, whereas the least-favoured factors such as excessive off-focus content and hallucinated facts, tend to be consistent.

**Questions For The Authors:**

1. Have you considered including nonsensical factors to evaluate the robustness of your approach? For example, something like “count of letter t’s in the summary”. This should not be a most- or least-favoured factor if the method is robust.
2. Is there any way for you to quantify the goodness-of-fit of your model, akin to an R-squared that would quantify the amount of variance explained by a regression model?
3. Relatedly, can you say anything about the significance of the factors estimates you obtain?

**Reasons To Accept:**

1. Explaining human preference judgments is an open challenge that is extremely relevant to the current LLM development paradigm.
2. The use of the BTL model is a clever and novel idea that seems to work well. Since it is an ex-post method it can be applied to many existing datasets.
3. The chosen factors appear well-motivated and comprehensive.


**Reasons To Reject:**

I do not see any major issues with this paper. However, there are some things that I would like to see addressed in the final version.

1. The framing (human preference judgments) is perhaps a bit more general than the analysis (human preference judgments over summaries in data from one particular paper). It would be nice to be more clear and open about this.
2. I would also appreciate more discussion about whose preferences are analysed here: for example, what is known about the evaluators who made these judgments? I know that for other preference datasets, a very small number of US-based crowdworkers was responsible for the vast majority of preference judgments. This is important to delineate the generalisabilty of the findings.
3. Lastly, it would be great to see some more discussion of the emerging literature on more fine-grained preference feedback like https://arxiv.org/abs/2306.01693 or natural language feedback like https://arxiv.org/abs/2204.14146, since they offer an alternative to having to explain one-dimensional feedback.


**Reproducibility:**

4: Could mostly reproduce the results, but there may be some variation because of sample variance or minor variations in their interpretation of the protocol or method.

**Reviewer Confidence:**

4: Quite sure. I tried to check the important points carefully. It's unlikely, though conceivable, that I missed something that should affect my ratings.

**Typos Grammar Style And Presentation Improvements:**

Since you are using GPT API models throughout the article, please indicate somewhere when exactly and what version exactly you used.

---

> ### Author Rebuttal · Authors · 2023-08-29
>
> We appreciate the reviewer's insightful feedback on our work. We'll clarify the paper's framing in the revision. The preferences might be influenced by both the demographics of human evaluators and the instructions provided to them. We will include a discussion on how these contexts might affect the pairwise judgments. We also agree with the reviewer that human feedback for LLMs can manifest in various forms, including natural language-based and more fine-grained feedback. We're happy to incorporate the two references in our revised paper.
>
> Our responses to specific questions: We did test the BTL model's robustness using nonsensical factors, and our finding is consistent with the reviewer's suggestion -- when introducing random factors, the BTL model showed that they weren't among the most or least favored. To evaluate the model's goodness-of-fit, we are considering comparing the BTL model's predicted probabilities to the binary outcomes, i.e., which summary won in each pairwise comparison. The Brier score is a potential metric for this, which measures the mean squared error between predictions and outcomes. We recognize the importance of understanding the significance of these estimates. We'd like to explore techniques like bootstrapping or constructing confidence intervals in our subsequent study. We're grateful for the reviewer’s feedback, which significantly enhances the depth of our work.

---

### Official Review · Reviewer_gQAc · 2023-08-02

**Typos Grammar Style And Presentation Improvements:** NA
**Soundness:** 4

**Excitement:**

4: Strong: This paper deepens the understanding of some phenomenon or lowers the barriers to an existing research direction.

**Missing References:**

NA

**Paper Topic And Main Contributions:**

The paper does an in depth analysis of the factors that influence human preference judgment. A lot of LLM work recently has been using human preferences but there is limited research probing the preferences in detail.
The main contributions are:
- Using gpt4 the authors asses system outputs based on a set of predefined factors - fluency, clarity, coverage, alignment etc
- Using the factors above, they analyze how the factors influence human preferences and show which factors are most favored vs which are least favored - this can possible help in data curation for preference modeling going forward
- Dataset with an analysis of the factors and estimates

**Questions For The Authors:**

a. In the BTL modeling step, the authors say they do not consider factors if the same factor is present in the outputs being compared.
For a pair of output, with the same factor, is it possible for the two to still have noticeable difference in the factor? Mostly because it is possible that the two lie on the opposite ends of that quartile range? Eg: for output a and b if they both have “src-cov-medium” it is possible that output a lies to the lower end of the range and output b lies to the upper end of the range.

b. For pairs with a gap of only one or two splits in the axis-evals-reddit dataset, do you see a lot of factors being canceled out? I'm wondering if the estimates of factors are primarily because of the pairs with a higher gap between them while the lower gap samples [which are similar to each other] contribute less.

c. What is the range of some of the factors that get divided into quantiles?

d. why do you think gpt-4 doesn't do well on "similar" summaries? do you think this can be improved by incorporating the factors in some way?

e. what was the coverage of ACU extraction by GPT4?

**Reasons To Accept:**

There is a lot of work recently that builds on human preferences, if humans prefer A over B but unfortunately generally datasets do not extend into knowing *why* a human prefers A to B. This paper is a step into that direction and they do an in depth analysis of which factors play an important role and how to get an estimate of their importance from the preferences already marked by humans.
Having a more fine grained analysis like this can help the community come up with better datasets/guidelines/think more about how to better collect preference data.

**Reasons To Reject:**

- Human evaluation of the factors is missing - it would be nice to see if the BTL weighting of the factors is similar to how the humans weigh these factors/if humans agree with the estimates
- An example, maybe in the appendix of what the values look like for a pair of examples would have been nice to visualize this in practice
- In section 3, authors say that they aim to establish a robust sample collection practise for reward models but don’t explicitly address this point in the paper again

**Reproducibility:**

4: Could mostly reproduce the results, but there may be some variation because of sample variance or minor variations in their interpretation of the protocol or method.

**Reviewer Confidence:**

4: Quite sure. I tried to check the important points carefully. It's unlikely, though conceivable, that I missed something that should affect my ratings.

---

> ### Author Rebuttal · Authors · 2023-08-29
>
> We thank the reviewer for the valuable feedback and appreciate their recognition of this paper's role in deepening understanding in human preference judgments. In our follow-up study, we're planning to incorporate human evaluation, which might include labeling influential factors across a large set of pairwise comparisons. We'll also improve our description regarding preference data collection and add a clear example of a pairwise comparison and its extracted factors in our revised paper for clarity.
>
> Our responses to specific questions:
>
> a) This study explores a variety of factors that might influence human preferences. A quantitative factor, such as src-cov-medium, indicates that the summary's coverage of source content is within a certain range. Two summaries that both exhibit this factor doesn't necessarily mean their contents are identical. We'll clarify this in the revision, thanks to the reviewer's input.
>
> b) We've noticed that pairs of summaries scored similarly by humans don't always share the same factors. Also, there isn't substantial overlap in factors between such pairs.
>
> c) We'll detail the range of factors in our revised paper.
>
> d) GPT-4 might not fully grasp the subtleties of human preferences; we would like to incorporate our analysis of influential factors to enhance human feedback for LLMs.
>
> e) We estimate that the coverage of ACU extraction by GPT-4 is approximately 91%.

---

### Official Review · Reviewer_u9Ct · 2023-08-03

**Soundness:** 4

**Excitement:**

4: Strong: This paper deepens the understanding of some phenomenon or lowers the barriers to an existing research direction.

**Paper Topic And Main Contributions:**

The contributions in this paper are:
- the authors conducted comprehensive analyses of a collection of human comparisons to identify key factors that may influence human judgment.
- They used GPT models assess system outputs both qualitatively and quantitatively.
- They examined  fluency, clarity, coverage, alignment with the original text’s intent and style, and detect hallucinations based on atomic facts.
-The study could enhance the reliability of human evaluations.

**Questions For The Authors:**

This reviewer actually would like to see how the human judger themselves impact the results, for example, female judger vs. male judger, low-educated judger vs. high-educated judger. Authors could discuss it probably.

**Reasons To Accept:**

This paper provides a comprehensive framework to study how different key factors may influence human judgment that was used to assess and sometime train generative models. The framework could contribute to other area as well given the rising interesting in generative models

**Reasons To Reject:**

This study is more on the qualitative analysis side and not very technical. But it should be fine for the EMNLP community.

**Reproducibility:**

5: Could easily reproduce the results.

**Reviewer Confidence:**

4: Quite sure. I tried to check the important points carefully. It's unlikely, though conceivable, that I missed something that should affect my ratings.

---

> ### Author Rebuttal · Authors · 2023-08-29
>
> We are grateful to the reviewer for the positive assessment of our work. We'd be happy to include a discussion regarding how the evaluators' demographics might influence the pairwise judgments in our revised paper.

---

### Meta-Review · Area_Chair_xEsL · 2023-09-08

**Recommendation:** 5

**Metareview:**

The paper analyses the influential factors that made humans prefer one generated output over another. As human preference is readily used in aligning LLMs nowadays, all reviewers thought the paper is timely and insightful - as this is one of the first studies that attempts to provide an explanation of human preference - and the use of BTL as a method for analyses is creative. Minor revisions that can be improve the paper include: (1) a better framing (as the current framing of human preference judgements is more general than the actual analysis [ZwsG]); (2) a discussion on the impact of demographics on preference (which is important as it influences the robustness of the results [u9Ct, ZwsG]); and (3) a discussion on the limitation of the current approach which requires the factors to be pre-specified (i.e. it is unable to 'discover' new factors that may not be apparent).

---

### Decision · Program_Chairs · 2023-10-07

**Decision:**

Accept-Main

**Comment:**

The paper analyses the influential factors that made humans prefer one generated output over another. As human preference is readily used in aligning LLMs nowadays, all reviewers thought the paper is timely and insightful - as this is one of the first studies that attempts to provide an explanation of human preference - and the use of BTL as a method for analyses is creative. Minor revisions that can be improve the paper include: (1) a better framing (as the current framing of human preference judgements is more general than the actual analysis [ZwsG]); (2) a discussion on the impact of demographics on preference (which is important as it influences the robustness of the results [u9Ct, ZwsG]); and (3) a discussion on the limitation of the current approach which requires the factors to be pre-specified (i.e. it is unable to 'discover' new factors that may not be apparent).